# Genome-wide DNA methylation is predictive of outcome in juvenile myelomonocytic leukemia

Elliot Stieglitz [1,2], Tali Mazor[3], Adam B. Olshen[2,4], Huimin Geng[5], Laura C. Gelston[1], Jon Akutagawa[1], Daniel B. Lipka [6,7,8], Christoph Plass[6,9], Christian Flotho[9,10], Farid F. Chehab[11], Benjamin S. Braun[1,2], Joseph F. Costello[3] & Mignon L. Loh[1,2]

Juvenile myelomonocytic leukemia (JMML) is a myeloproliferative disorder of childhood caused by mutations in the Ras pathway. Outcomes in JMML vary markedly from spontaneous resolution to rapid relapse after hematopoietic stem cell transplantation. Here, we hypothesized that DNA methylation patterns would help predict disease outcome and therefore performed genome-wide DNA methylation profiling in a cohort of 39 patients. Unsupervised hierarchical clustering identifies three clusters of patients. Importantly, these clusters differ significantly in terms of 4-year event-free survival, with the lowest methylation cluster having the highest rates of survival. These findings were validated in an independent cohort of 40 patients. Notably, all but one of 14 patients experiencing spontaneous resolution cluster together and closer to 22 healthy controls than to other JMML cases. Thus, we show that DNA methylation patterns in JMML are predictive of outcome and can identify the patients most likely to experience spontaneous resolution.

[1] Department of Pediatrics, Benioff Children's Hospital, University of California, San Francisco, 1450 3rd Street, San Francisco, CA 94158, USA. [2] Helen Diller Family Comprehensive Cancer Center, University of California, San Francisco, 1450 3rd Street, San Francisco, CA 94158, USA. [3] Department of Neurological Surgery, University of California, San Francisco, 1450 3rd Street, Room 471, San Francisco, CA 94158, USA. [4] Department of Epidemiology and Biostatistics, University of California, San Francisco, 550 16th Street, Box 0560, San Francisco, CA 94158, USA. [5] Departments of Laboratory Medicine and Cellular and Molecular Pharmacology, University of California, 513 Parnassus Avenue, 1457A, Box 0451, San Francisco, CA 94143, USA. [6] Division of Epigenomics and Cancer Risk Factors, German Cancer Research Center (DKFZ), Im Neuenheimer Feld 280, 69120 Heidelberg, Germany. [7] Department of Hematology and Oncology, Medical Center, Otto-von-Guericke-University, Leipziger Str. 44, 39120 Magdeburg, Germany. [8] Health Campus Immunology, Infectiology and Inflammation, Otto-von-Guericke-University, Leipziger Str. 44, 39120 Magdeburg, Germany. [9] German Cancer Consortium (DKTK), Im Neuenheimer Feld 280, 69120 Heidelberg, Germany. [10] Division of Pediatric Hematology and Oncology, Department of Pediatrics and Adolescent Medicine, Medical Center, Faculty of Medicine, University of Freiburg, Hugstetter Str. 55, 79106 Freiburg, Germany. [11] Department of Laboratory Medicine, University of California, San Francisco, 185 Berry Street Bldg B 290, Box 0134, San Francisco, CA 94158, USA. Elliot Stieglitz, Tali Mazor and Adam B. Olshen contributed equally to this work. Correspondence and requests for materials should be addressed to E.S. (email: elliot.stieglitz@ucsf.edu) or to M.L.L. (email: mignon.loh@ucsf.edu)

Juvenile myelomonocytic leukemia (JMML) is a rare and aggressive myeloproliferative disease of childhood associated with heterogeneous outcomes[1, 2]. JMML is initiated by mutations in Ras pathway genes that lead to hyperactive Ras signaling[3]. Several recent reports have identified secondary mutations in epigenetic-regulating genes, including members of the polycomb repressor complex 2 (PRC2), whose presence correlates with higher rates of relapse[4–6]. In addition, *SETBP1* is mutated in up to 30% of patients as a secondary event, and we previously showed that subclonal mutations in *SETBP1* at diagnosis are associated with poor outcome[7].

Spontaneous resolution of JMML with minimal to no traditional chemotherapy or hematopoietic stem cell transplantation (HSCT) is rare but has been reported[8]. Although this phenomenon is more common in patients with germline syndromes such as Noonan[9] and CBL syndrome[10], it has also been seen in a handful of patients with somatic *NRAS* and *KRAS* mutations[8] that are typically less than one year of age and have high platelet counts. Interestingly, spontaneous resolution does not commonly occur in patients with Neurofibromatosis type I, a common congenital condition with a high incidence of JMML. In general, robust biomarkers to predict spontaneous resolution are still lacking.

We know that patients with identical somatic Ras pathway mutations have divergent outcomes[5, 6]. We hypothesized that either the cell in which genetic mutations arise or potentially non-genetic changes such as DNA methylation could explain differences in outcome[11]. Considering that DNA methylation has a critical role in the differentiation of normal fetal and adult hematopoietic stem cells[12–14], we suspected that leukemogenesis in JMML is in part determined by alterations in the JMML methylome, and that such differences may predict outcome. Importantly, prior reports investigating the role of DNA methylation in this leukemia were limited to a handful of candidate genes previously shown to be altered in other myeloid malignancies[15, 16]. We therefore sought an unbiased, genome-wide approach to define the DNA methylome of newly diagnosed JMML patients and then evaluated whether a specific DNA methylation signature was capable of predicting outcome with a particular emphasis on identifying patients who experienced spontaneous resolution.

In this study, we generated genome-wide DNA methylation data in a discovery cohort of 39 patients with JMML and validated our findings in an independent cohort of 40 additional patients. We found that JMML patients cluster into three subgroups that were independently predictive of outcome based on their methylomes. The low methylation subgroup was associated with high rates of survival and spontaneous resolution, whereas the high methylation subgroup was associated with dismal survival. These data suggest that DNA methylation can be used to predict outcome in JMML.

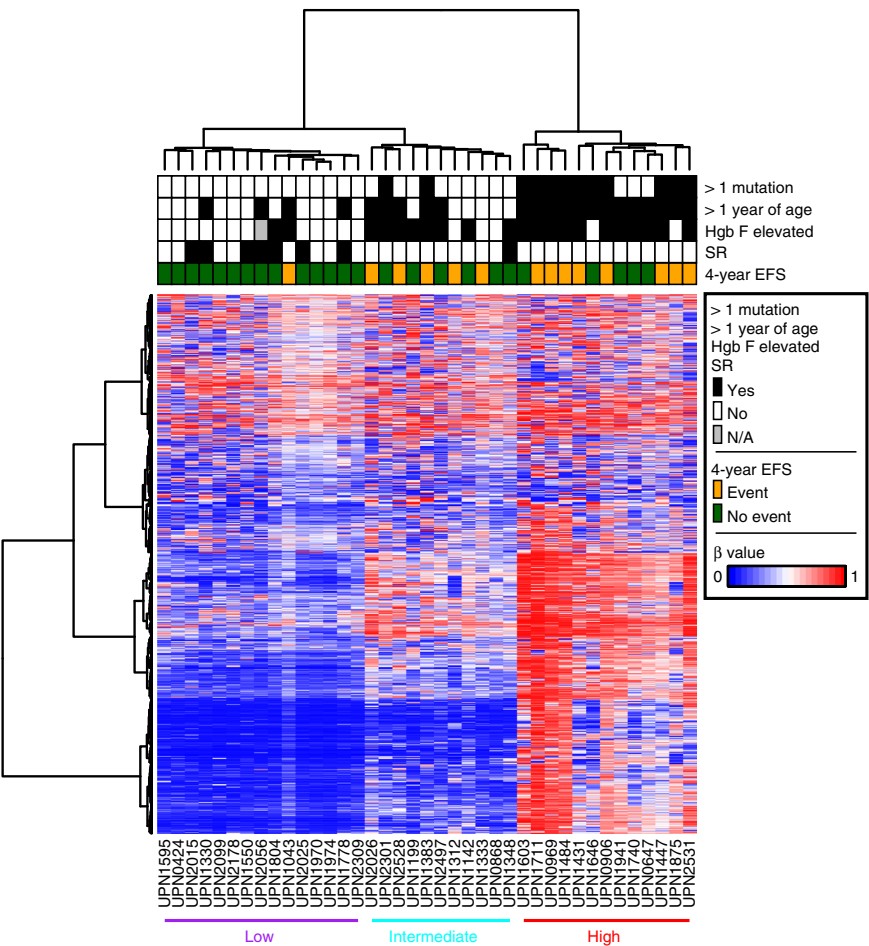

**Fig. 1** Unsupervised clustering reveals three distinct clusters of DNA methylation in patients with JMML. Thirty-nine patients who underwent Illumina 450k analysis are included. Patients are displayed on the X axis and the 1527 most variable CpG sites (top 0.5% ranked by standard deviation) are displayed on the Y axis. The three most significant patient characteristics in univariable analysis are presented at the top of the figure. HgB F fetal hemoglobin, SR spontaneous resolution, EFS event-free survival, β beta value, N/A not available

## Results

**Genetic mutations are present in putative HSCs.** We sorted samples from four patients to determine whether the presence or absence of Ras driver mutations in putative hematopoietic stem cells could explain the phenomenon of spontaneous resolution. Individual patient samples were sorted into early hematopoietic stem cells, multipotent progenitors, and common myeloid progenitors. The Ras pathway mutation was present in each compartment in all four patients including the two patients that went on to experience spontaneous resolution (Supplementary Table 1). We therefore pursued the alternative theory that DNA methylation influences outcome.

**Pilot study reveals minimal variation in DNA methylation.** We first sought to quantify the differences in both mutational burden and DNA methylation in $CD14^+$ and unselected mononuclear cell DNA. $CD14^+$ cells would be most likely to be enriched with a leukemic signature due to the monocytic nature of the disease. To determine the allelic fraction of somatic Ras pathway mutations in different cell lineages we performed deep sequencing of individual mutations in $CD3^+$, $CD14^+$, $CD19^+$, and $CD34^+$ compartments and observed nearly identical allelic fractions in $CD14^+$ cells compared to unselected mononuclear cell DNA (Supplementary Table 2). All four cell lineages were found to harbor the Ras pathway mutation with varying allelic fractions, although lymphoid cells including $CD3^+$ and $CD19^+$ populations had the lowest mutational burden (Supplementary Table 2). We next performed enhanced reduced representation bisulfite sequencing (eRRBS) on $CD14^+$ and unselected mononuclear cell DNA from the same three JMML patients as well as three age-appropriate healthy controls. DNA from unselected mononuclear cells and $CD14^+$ cells from each subject had very similar DNA methylation levels (concordance correlation coefficient range: 0.98 for each subject) (Supplementary Fig. 1a, b). In an unsupervised clustering analysis, $CD14^+$ samples also clustered with the unselected mononuclear cell sample from the same subject as opposed to $CD14^+$ samples from other subjects (Supplementary Fig. 1c). We therefore used unselected mononuclear cell DNA for all ensuing experiments.

**JMML patients have three distinct clusters of methylation.** We next generated Illumina 450k methylation data for a discovery cohort of 39 well-characterized JMML patients using unselected mononuclear cell DNA (Supplementary Table 3). We performed an unsupervised clustering analysis based on the 1527 most variably methylated CpG sites. We observed an inverse relationship between 4-year event-free survival (EFS) and the degree of DNA methylation (Fig. 1). For patients in the cluster with the lowest levels of DNA methylation in the selected 1527 most variable CpG sites, the proportion of patients having an event at 4 years was 6% (1/15; 95% confidence interval (CI), 2%-32%). This compared to 45% (5/11; CI: 17–77%) for patients in the cluster of intermediate levels of methylation and 61% (8/13; CI: 32–86%) for those patients with the highest level of methylation (Supplementary Fig. 2). The proportion of patients with events differed significantly by cluster ($P = 0.0039$).

**Secondary mutations are associated with hypermethylation.** Several groups have recently demonstrated that secondary mutations are associated with a worse prognosis in JMML[4–6]. We observed that all patients who had a secondary mutation present at diagnosis were classified in either the intermediate or high methylation cluster. The secondary genetic mutations occurred in genes affecting DNA methylation (ASXL1, DNMT3A) as well as Ras pathway genes (NRAS, NF1), spliceosome members (ZRSR2),

**Table 1 Univariable regression analysis**

| Univariable analysis | EFS from date of diagnosis | | | |
|---|---|---|---|---|
| | **N** | **OR** | **95% CI** | **P-value** |
| Age at diagnosis (months) | | | | 0.004 |
| ≤12 months | 17 | 1 | | |
| >12 months | 22 | 9 | 1.93–66.41 | |
| Methylation cluster | | | | 0.0039[a] |
| Low | 15 | 1 | | |
| Intermediate | 11 | 11.67 | 1.48–250.68 | |
| High | 13 | 22.4 | 3.05–474.51 | |
| Somatic mutations at diagnosis | | | | 0.0007 |
| ≤1 | 27 | 1 | | |
| >1 | 12 | 13.2 | 2.86–79.13 | |
| HbF at diagnosis | | | | 0.024 |
| Not elevated for age | 17 | 1 | | |
| Elevated for age | 21 | 5.13 | 1.23–27.33 | |
| Platelet count at diagnosis ×10⁹ | | | | 0.29 |
| ≤50 | 17 | 1 | | |
| >50 | 20 | 0.48 | 0.12–1.84 | |
| Somatic PTPN11 mutation | | | | 0.1 |
| No | 26 | 1 | | |
| Yes | 13 | 3.17 | 0.80–13.39 | |
| Gender | | | | 0.97 |
| Male | 28 | 1 | | |
| Female | 11 | 1.029 | 0.22–4.32 | |

transcription factors (SH2B3, GATA2), and SETBP1 (Supplementary Table 4).

**Methylation status is predictive of outcome.** In univariable regression analyses (Table 1), the characteristics that reached significance at the 0.05 level for 4-year EFS were age at diagnosis of ≥12 months (OR = 9, CI = 1.93–66.41, $P = 0.0040$), somatic mutations >1 (OR = 13.2, CI = 2.86–79.13, $P = 0.0007$) elevated fetal hemoglobin (OR = 5.13, CI = 1.232-27.33, $P = 0.024$), and methylation cluster ($P = 0.0039$). Furthermore, when a multivariable regression model was applied using the three variables with a univariable $P < 0.01$ (methylation cluster group, somatic mutations, and age), the methylation cluster grouping retained statistical significance for 4-year EFS ($P = 0.032$) (Supplementary Table 5). The number of somatic mutations was also significant ($P = 0.018$), whereas age was not ($P = 0.12$).

**Independent cohort validates prognostic value of methylation.** We next sought to evaluate our findings in an independent cohort of patients. We obtained Illumina 450k data from 40 JMML patients treated in the EWOG-MDS consortium, all of whom met the international criteria for JMML[3], were negative for Noonan syndrome, had 4-year EFS data available, and did not experience treatment related mortality. Thirty-three patients underwent HSCT and the remaining seven patients experienced spontaneous resolution without HSCT (Supplementary Table 6). Hierarchical clustering of these samples using the set of 1527 CpG sites defined in our discovery cohort revealed a similar pattern of three clusters, varying from low to intermediate to high methylation levels (Fig. 2). Moreover, all seven patients in the EWOG-MDS cohort that experienced spontaneous resolution clustered together in the lowest methylation cluster.

We classified the validation cohort into methylation groups based on data from the discovery cohort. Specifically, we assigned each validation patient as low, intermediate or high, based on the shortest distance to the corresponding discovery cohort cluster centroid. The proportion of patients classified to the low methylation cluster having an event at 4 years was 8% (1/12;

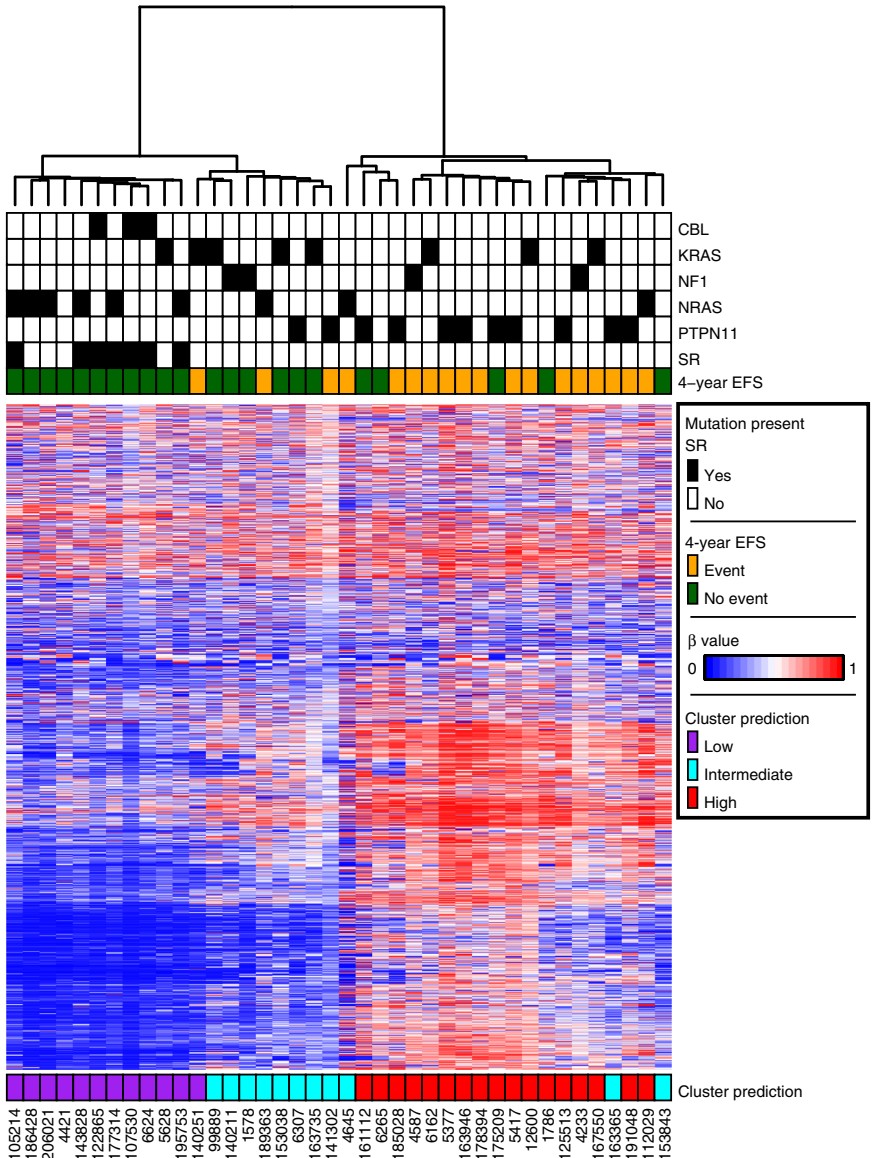

**Fig. 2** DNA methylation predicts outcome in an independent cohort. Forty patients from EWOG-MDS were analyzed. Patients are displayed on the X axis and the same 1527 most variable CpG sites from the unsupervised discovery cohort analysis are displayed on the Y axis. Patients are assigned a cluster designation based on minimum distance to the centroid of the discovery cohort clusters. HgB F fetal hemoglobin, SR spontaneous remission, EFS event-free survival, β beta value

CI: 0–38%). This compared to 36% (4/11; CI: 11–69%) for patients classified to the cluster of intermediate levels of methylation and 76% (13/17; CI: 50–93%) for those patients classified to the highest level of methylation. The proportion of patients with events differed significantly by cluster ($P = 0.0008$) (Fig. 2).

**Methylation changes are consistent across tissue types**. Tissue sources for DNA in our discovery and validation cohorts included mononuclear cells or granulocytes derived from bone marrow ($n = 27$), peripheral blood ($n = 39$), and spleen ($n = 13$). Tissue source did not appear to influence the clustering designation of low, intermediate or high (Supplementary Fig. 3). A Fisher's exact test relating tissue source and cluster membership had a $P$-value of 0.45.

**Spontaneous remitters cluster with healthy controls**. We then compared our combined cohort of 79 JMML patients with 22

healthy, age-appropriate controls analyzed on the Illumina 450k platform using peripheral blood mononuclear cell DNA. Notably, using the same set of 1527 CpG sites defined by the discovery cohort, 27/79 JMML patients clustered more closely to the 22 healthy controls than the other JMML cases (Fig. 3). Of these 27 patients, 14 (52%) experienced spontaneous resolution and only two (7%) experienced an event within 4 years.

**Identification of a methylation signature to predict outcome**. In preparation for implementing a clinical test we sought to identify the fewest number of CpG probes that would be necessary to recapitulate the low, intermediate and high methylation clusters from our discovery cohort. For this purpose, we used the discovery cohort to rank probes and the cluster call of the validation cohort to assess accuracy. Disagreements with the calls based on all 1527 probes were considered errors. With 50 or greater probes, the number of errors went down to either 0 or 1 (Supplementary Fig. 4).

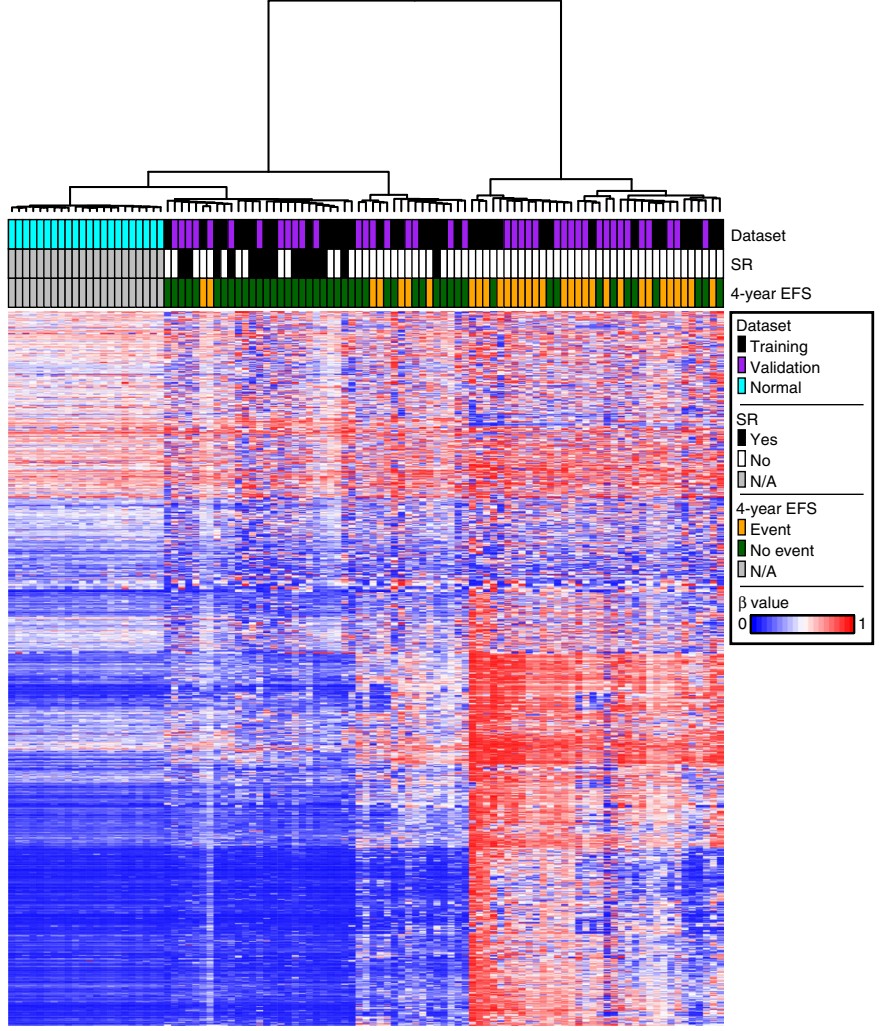

**Fig. 3** Patients experiencing spontaneous resolution cluster closer to healthy age-appropriate controls. Twenty-two healthy, age-appropriate controls were analyzed together with the 79 JMML patients from the combined discovery and validation cohorts. Patients and controls are displayed on the X axis and the same 1527 most variable CpG sites from the unsupervised discovery cohort analysis are displayed on the Y axis. HgB F fetal hemoglobin, SR spontaneous resolution, EFS event-free survival, β beta value, N/A not applicable

**Impact of DNA methylation signature on expression**. We examined the interplay between altered DNA methylation in the most variable 1527 CpG probes and gene expression. To perform this analysis, we utilized the subset of those 1527 CpG sites that were in gene promoters to generate a list of differentially methylated genes and performed unsupervised clustering based on the expression of those genes. The samples did not cluster into the previously identified low, intermediate, and high methylation classes (Supplementary Fig. 5).

**Pathway analysis reveals enrichment in the Ras/MAPK pathway**. We next carried out supervised analyses using the combined discovery and validation cohorts to identify CpG sites that differed between the 32 patients who experienced events and the 47 patients who did not. We identified 10,545 significantly differentially methylated CpG sites. To focus on those CpG sites most likely to change gene expression, we retained only those CpG sites that were in gene promoters, yielding a list of 3419 genes (Supplementary Fig. 6). KEGG pathway analysis of these same genes revealed an enrichment in several pathways known to be relevant in initiating JMML, including MAPK, PI3K-AKT, and Ras signaling, implicating these variably methylated CpG sites in leukemogenesis (Supplementary Table 7).

We next sought to use an empirical approach to identify CpG sites that were most likely to influence changes in gene expression. We analyzed RNA-Seq files of eight patients from our discovery cohort and performed a supervised analysis to identify the most variably expressed genes that differed between patients who experienced events within 4 years compared to those who did not. We then looked at the overlapping genes from the supervised analyses using both the DNA methylation and RNA-Seq data and identified 43 unique genes. KEGG pathway analysis identified transcriptional misregulation in cancer, Ras signaling pathway, and pathways in cancer as the top three results (Supplementary Table 8).

**Discussion**

JMML is an aggressive myeloproliferative disorder predominantly affecting infants and toddlers. It is considered one of the purest forms of a Ras-driven leukemia with ~95% of patients harboring at least one mutation in the Ras pathway. Several groups have now reported that secondary genetic mutations within and outside of the canonically defined Ras pathway are present in roughly a third of patients and that these mutations are associated with a worse prognosis[4–6]. Even with this knowledge, identifying patients in need of more intensive therapy at diagnosis is still a

challenge for providers. Equally difficult is predicting the phenomenon of spontaneous resolution frequently experienced by patients with Noonan or CBL syndrome, and occasionally in young patients with somatic *NRAS* and *KRAS* mutations.

One proposed model for understanding the divergent outcomes in this disease centers around the putative initiating hematopoietic cell in which the causative Ras mutations first arise. We first hypothesized that patients who experience spontaneous resolution would have somatic mutations only in late, committed progenitors as opposed to patients with aggressive disease who possibly have mutations occurring in earlier stem cells. For example, we have previously demonstrated that *SETBP1* mutations occur in early progenitor cells including CD34$^+$/CD45RA$^-$/CD90$^+$ (putative HSCs)[7]. Here we show that patients who experienced spontaneous resolution also had somatic mutations in HSCs and thus the cell of origin in which genetic mutations arise are unlikely to explain differences in outcome. We therefore explored epigenetic changes, specifically DNA methylation, as a possible explanation for divergent outcomes.

Using our discovery cohort, we found that JMML patients suffering from aggressive disease have a distinctly hypermethylated DNA profile at the most variable CpG sites compared to healthy controls in the same age range. In contrast, nearly half of JMML patients clustered more closely with healthy controls who were of a similar age range than with other patients suffering from aggressive disease. All but one of these patients who experienced spontaneous resolution clustered with healthy controls and we thus propose that further developing this methylation signature into a Clinical Laboratory Improvement Amendments/College of American Pathologist (CLIA/CAP) approved assay could lead to the first biomarker capable of predicting a milder disease course not requiring HSCT. In preparation for implementing a clinical assay, we have agreed to pool all publicly available DNA methylation data in this disease and develop a single sequencing-based assay across multiple continents. Of note, this pattern was consistent in bone marrow, peripheral blood, and spleen, suggesting that a fundamental mechanism underpins the DNA methylation signature in all of the infiltrating cells involved in this disease.

Pathway analysis confirmed that the most variably methylated CpG sites that were used in our supervised analysis were centered on genes involved in development and Ras/MAPK/PI3K/AKT signaling. However, after analyzing a small but representative subset of our JMML patients at the gene expression level, we believe that altered DNA methylation in JMML is functioning in a more complex manner than simply altering expression of associated genes. Recent work has demonstrated that patients with more than one mutation in the Ras pathway have inferior event-free survival compared to patients with only one Ras pathway mutation[6]. Our analysis shows that patients with more than one mutation tend to display hypermethylated DNA; however, the relationship between the two is not yet clear. In several myeloid malignancies, alterations in DNA methylation are a result of genetic mutations in epigenetic-regulating genes[17]. Surprisingly, JMML patients with any secondary mutation inclusive of those affecting genes in the Ras pathway, transcription factors and the spliceosome machinery all have hypermethylated signatures, not just patients with secondary mutations in genes directly affecting DNA methylation. We therefore hypothesize that alterations in DNA methylation are an early event in JMML and are permissive for the acquisition of secondary mutations and not vice versa.

Several important questions remain, including what initially causes the hypermethylation seen in patients with JMML? Similarly, how does a hypermethylated signature contribute to the aggressive disease seen in these patients? In addition, what role if any will hypomethylating agents have in this disease, and are they capable of reversing the hypermethylated state observed at diagnosis?.

This study highlights the utility of DNA methylation as a potential biomarker that could be used in a combined risk stratification algorithm along with clinical characteristics including age, fetal hemoglobin, and number of somatic mutations at diagnosis in future clinical trials. For JMML patients that are classified into the lowest methylation group clinicians could now consider careful observation or treatment with low intensity regimens like azacytidine or 6-mercaptopurine. In contrast, patients with the highest levels of methylation have unacceptably poor outcomes even after HSCT, with 75% of these patients experiencing relapse of their disease post-transplant. These patients should receive novel treatments in the context of clinical trials even at initial diagnosis.

In summary, our results show the potential of DNA methylation as a biomarker that can both identify patients who are predicted to fail HSCT as well as those who are most likely to experience spontaneous resolution and could be observed to avoid the acute and late side effects of HSCT.

## Methods

**Patients**. Thirty-nine distinct JMML patients comprising our discovery cohort had samples available at diagnosis and were analyzed using the Illumina Infinium HumanMethylation450 BeadChip platform. These patients were well-characterized in terms of diagnostic variables, treatment delivered, and genetic mutation status (Supplementary Table 9). Three separate JMML patients and three healthy controls were included in a pilot eRRBS study (Supplementary Table 10) described below. A validation cohort using 40 patients enrolled in the prospective 98 and 2006 trials (www.clinicaltrials.gov: #NCT00047268, #NCT00662090) was provided by our colleagues from the European Working Group of MDS and JMML in Childhood (EWOG-MDS). The EWOG-MDS patients were treated on a single clinical trial and had Illumina Infinium HumanMethylation450 BeadChip data available. Lastly, HumanMethylation450 BeadChip data for peripheral blood mononuclear cells from 22 healthy subjects (1–5 years of age) was obtained from a previously published study[18]. Approval for these studies were obtained from the University of California San Francisco (UCSF) Committee on Human Research. All participants/guardians provided informed consent in accordance with the Declaration of Helsinki.

**eRRBS sample processing and library generation**. Mononuclear cells were obtained from bone marrow and peripheral blood from three JMML patients and three controls after Ficoll separation. CD3, CD14, CD19, and CD34 positive cells were selected using magnetic beads (Stemcell Technologies) and DNA was extracted for each population. Deep sequencing (~1300X) of the initiating mutation for each of the three JMML patients was then carried out in unselected mononuclear cell, CD3$^+$, CD14$^+$, CD19$^+$, and CD34$^+$ populations. Unselected mononuclear cell and CD14$^+$ DNA from each of the six subjects was then subjected to eRRBS by digesting with MspI followed by end-repair, A-tailing, and ligation of methylated adapters. GC-rich fragments were selected and converted with bisulfite prior to PCR amplification and sequencing[19].

**Analysis of eRRBS data**. The amplified libraries were sequenced on an Illumina Genome Analyzer II or HiSeq2000 per manufacturer's recommended protocol for 50 bp single end read runs. Image capture, analysis and base calling was performed using Illumina's CASAVA 1.7. eRRBS sequencing data was aligned to whole genome using the bismark alignment software[20] with a maximum of two mismatches in a directional manner and only uniquely aligning reads were retained. To call methylation score for a base position, we required that at least 10 reads cover the position with a phred base quality of at least 20. Only CpG dinucleotides that satisfy these coverage and quality criteria were retained for subsequent analysis. Percentage of bisulfite converted Cs (representing unmethylated Cs) and nonconverted Cs (representing methylated Cs) were recorded for each C position in a CpG context. We retained methylation calls for all CpG sites with at least 10 reads of coverage in all datasets (790,359 CpG sites). We then calculated the difference at each CpG site between DNA from unselected and CD14$^+$ cells for each patient, as well as the concordance correlation coefficient between the two datasets. Unsupervised hierarchical clustering was performed using the most variable CpG sites (standard deviation >30 = 3277 CpG sites) using Ward's method. Implementation was through the hclust function in the stats R package.

**HumanMethylation450 bead array sample processing**. DNA from JMML patients was extracted using standard methods from bone marrow, splenic tissue or peripheral blood mononuclear cells or granulocytes obtained at diagnosis. Genomic DNA was bisulfite converted using the EZ DNA Methylation Kit (Zymo Research) and processed on Infinium HumanMethylation450 bead arrays (Illumina Inc.) according to the manufacturer's protocol.

**HumanMethylation450 bead array data processing**. Raw data was processed using the minfi R package[21]. In particular, the functions preprocessNoob, mapToGenome, and ratioConvert were utilized in that order to produce methylation beta-values ($\beta$) and $M$-values for every probe and sample. Probes that mapped to regions with known germline polymorphisms (Illumina supplementary SNP list v1.2, downloaded 3 September, 2013), to multiple genomic loci[22], or to either sex chromosome, along with probes where the maximum p-value was greater than 0.01 for at least one sample, were filtered out. This left 289,731 probes for our primary analysis. Any gene for which a probe was present within the promoter region (1.5 kb upstream to 1 kb downstream of TSS, Gencode v19 gene annotations) was associated with that probe[23].

**Hierarchical clustering-based sample classification**. The beta-values were hierarchically clustered in both directions (samples and probes) utilizing an unsupervised approach that used the Ward's method. The clustering was limited to probes with standard deviations greater than 0.25 across the discovery rate samples, which resulted in 1527 probes being utilized. The validation cohort samples were classified into one of three classes based on minimum distance to the centroid of the discovery cohort clusters. Implementation was through the hclust function in the stats R package.

**Statistical analysis**. Our primary endpoint was the binary variable 4-year event-free survival. Only one patient had an event after 4 years, which led us to choose this cutoff. We estimated survival curves using the Kaplan–Meier method. Logistic regression was utilized to model the relationships between 4-year event-free survival and clinical and genomic variables and significance was based on the likelihood ratio test. The cutoff for significance was a $P$-value <0.05.

**Differential methylation based on 4-year EFS**. Differential methylation between groups was estimated using the Limma method[24] on the methylation $M$-values through the limma R package. Testing was performed on every probe that was part of the main analysis. Significance was defined as a $P$-value <0.05 and a difference in average methylation between the two groups of at least 0.1.

**CpG distillation for recapitulation of clusters**. We used the discovery cohort to rank probes and the cluster call of the validation cohort to assess accuracy. We ranked probes from most typical to least typical on the discovery cohort, where the determination of "typical" was based on the distance from the sample to the centroid of the corresponding cluster summed over all samples. We made calls on the validation cohort using 5 to 200 probes, adding the best probes 5 at a time, with distance to the discovery cohort centroid as the basis of the call. Disagreements of calls based only a subset of probes compared to all 1527 probes were considered errors.

**RNA-sequencing**. Eight patients from our discovery cohort (Supplementary Table 11) had RNA extracted, which was used to prepare mRNA libraries and sequenced on the Illumina HiSeq platform, resulting in paired 50-nt reads; resulting data were subjected to quality control[25]. Gene expression was quantified for transcripts corresponding to GENCODE v12 genes using transcript per million mapped reads (TPM) using an RSEM-based pipeline[26]. All analyses were performed on the log base 2 of TPM + 1.

**Differential gene expression based on 4-year EFS**. Differential gene expression between groups was estimated using the Limma method[24]. Testing was performed on every gene that had non-zero counts. Significance was defined as a $P$-value <0.05.

**Hierarchical clustering based on RNA-Seq data**. The most variable 1527 CpG probes used to cluster patients in our methylation analysis were annotated to a gene, based on the presence of the CpG site in the promoter of the gene (1500 bp upstream of TSS to 1000 bp downstream of TSS). We only included genes with high variability using a standard deviation cutoff of 2.5, which eliminated genes with low read counts, resulting in the retention of 65 genes. We then clustered patients based on their median centered expression values.

**Pathway analysis**. Pathway analysis was performed on the genes corresponding to the probes identified as significant in the Limma analysis. For a gene to be included, the probe had to be in a promoter region. KEGG analysis was performed using KEGG Mapper (http://www.genome.jp/kegg/mapper.html).

**Progenitor sorting**. Previously cryopreserved bone marrow samples were thawed and sorted into the following fractions as described[27]: Lin⁻ (CD2, CD3, CD4, CD4, CD7, CD8, CD10, CD11b, CD14, C19, CD20, CD56, CD235a), CD34⁺ CD38⁻ CD45RA⁻CD90⁺ (hematopoietic stem cells), Lin⁻CD34⁺ CD38⁻ CD45RA⁻CD90⁻ (multipotent progenitors), Lin⁻CD34⁺ CD38⁺ CD45RA⁻ CD90⁻ (common myeloid progenitors), and Lin⁻CD34⁺ CD38⁺ CD45RA⁺ CD90⁻ (granulocyte–monocyte progenitors). Individual cells from a single compartment were then pooled together and DNA was extracted from pooled cells and analyzed by Sanger sequencing.

**Data availability**. The data used in this study are available for download from the European Genome-phenome Archive (EGA) repository under accession EGAS00001002700. All relevant data are available from the authors upon request.

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

## Acknowledgements

This work was supported by the National Institutes of Health, National Cancer Institute grants T32CA128583 (E.S.), R01CA173085 (M.L.L.), and 5P30CA082103 (A.B.O); National Institutes of Health, National Heart, Lung, and Blood Institute grant K08HL135434 (E.S.); the Rally Foundation for Childhood Cancer Research (E.S.); the Leukemia and Lymphoma Society (Grant Numbers 6059-09 and 6466-15) (M.L.L.); Alex's Lemonade Stand Foundation, Center of Excellence (E.S); the St. Baldrick's Foundation (E.S.); the Frank A. Campini Foundation (E.S. and M.L.L.); Hyundai Hope on Wheels (M.L.L.). This work was supported in part by funding from the German José Carreras Leukemia Foundation (DJCLS) to C.P., D.B.L., and C.F. (project: DJCLS R 15/01). M.L.L. is the UCSF Benioff Chair of Children's Health and the Deborah and Arthur Ablin Endowed Professor of Pediatric Molecular Oncology.

## Author contributions

E.S., T.M., L.C.G., and J.A. performed the experiments. E.S., T.M., A.B.O., H.G., and F.F. C. performed data analysis. D.B.L., C.P., and C.F., contributed reagents and materials. E.S., T.M., A.B.O., and M.L.L. wrote the first draft of the manuscript. A.B.O. performed statistical analysis. B.S.B., J.F.C., and M.L.L. supervised research. All co-authors contributed to the final version of the manuscript.

## Additional information

**Competing interests:** The authors declare no competing financial interests.

