## [Peer Review File · Nature Communications]

Reviewers' comments:

Reviewer #1 Expert in leukaemia genetics:

The authors study the methylation patterns of DNA of patients with juvenile myelomonocytic leukemia (JMML). The rationale to look at methylation was to try to better predict patients who do not need treatment and those who are in urgent need for treatment. Outcomes in this disease vary from spontaneous resolution to rapid relapse after hematopoietic stem cell transplantation. Genome-wide DNA methylation profiling was performed using the Illumina 450k platform in a discovery cohort of 39 patients and in an independent cohort of 40 patients, and compared to 22 healthy controls. Unsupervised hierarchical clustering based on the variable CpG sites identified three clusters of patients (high, medium and low methylation), with the lowest methylation cluster having the highest rates of survival. The authors nicely demonstrate that the cell type analysed does not influence the results (mononuclear blood cells or sorted CD14+ cells gave nearly the same methylation profiles). High methylation was predominantly found in patients with >1 mutation, older than 1 year, with HgB F elevated.

This is a strong study that shows that DNA methylation patterns in JMML are predictive of outcome and can identify patients who are most likely to experience spontaneous resolution of disease. However, this is the only message, no insight in the causes or consequences of altered DNA methylation is provided and no link with gene expression data is given.

Major remark:

- The DNA methylation profiles are clear, but have not been linked with expression data. Since DNA methylation is expected to influence expression, it would be a nice and essential addition to include also RNA-seq data for example on CD14+ sorted cells from a selection of patients with low, intermediate and high methylation. This would link the observation of DNA methylation changes to effects on gene expression that could explain the difference in outcome. Now the paper remains very short and descriptive. RNA-seq analysis will allow to determine if genes from specific pathways are more/less expressed, stem cell signatures are more present, identify escape from immune surveillance, etc.
- If gene expression would be used instead of methylation data, would similar clusters be identified? In other words: how different is methylation data from gene expression data for clustering of the patients?

Minor remarks:

- On line 156 it is described a multivariable regression model was used. Could you specify more in detail the model that was used?
- Patient data was collected from the EWOG-MDS consortium. Could you specify from which trials these data was?
- Which software/package/tool was used for the pathway analyses?

Reviewer #2 Expert in leukaemia epigenetics:

In the present manuscript, the authors hypothesized that DNA methylome may help predicting JMML outcome.

Genome-wide DNA methylation profiling using the 450k platform in a cohort of 39 patients was performed. The authors identified three clusters of patients, differing significantly in terms of 4-year event-free survival, with the lowest methylation cluster having the highest rates of survival.

This study demonstrates that DNA methylation patterns in JMML are predictive of outcome in this heterogeneously behaving disease and can identify patients who are most likely to experience spontaneous resolution.

1. A cohort of 79 JMML patients was compared with 22 healthy controls, using the set of 1,527 CpG sites defined by the discovery cohort, 27/79 JMML clustered closely to the 22 healthy controls (Fig. 3). Of these 27 patients, 14 (52%) experienced spontaneous resolution and only two (7%) experienced an event within 4 years.

Which is the EFS of the other 9 patients that also cluster with the normal and correspond to 41%?

2. The authors carried out supervised analyses using the combined discovery and validation cohorts to identify CpG sites that differed between the 32 patients who experienced events and the 47 that did not experience events. 881 significantly differentially methylated CpG sites were defined. The authors retained only those CpG sites that are in gene promoters, yielding a list of 633 genes. On the contrary, if retaining those CpG sites that do not stand in promoters, where and how those sites could be grouped? Would also this signature be important for disease progression and development? In addition, KEGG pathway analysis of these same 633 genes revealed enrichment in several pathways, including MAPK, PI3K-AKT, Rap1 and Ras signaling, implicating these variably methylated CpG sites in leukemogenesis. Were the differential methylations at these sites influencing the corresponding gene expression? If so would these data indicate a potential for targeting MAPK, PI3K-AKT in a subset of JMML?

3. Pathway analysis confirmed that the most variably methylated CpG sites that were used in the supervised analysis were centered on genes involved in development and Ras/MAPK/PI3K/AKT signaling. Was the variation relative to common hyper or hypomethylation at these sites, or?

4. The authors indicate that second mutations were found on some genes such as DNMT3. Which are the expression levels of those targets? Are they also affected by DNA hyper-hypomethylation? On a different corner, which is the mutated-DNMT3 enzymatic function Is it increasing or decreasing?

5. The reason why the DNA methylation is high or low in the different patients, is not clear and not experimentally addressed. It would be good to tune this down or include more data suggesting potential hypothesis.

6. The potential use of DNA demethylating agents should be investigated on primary cells or cell models with similar methylation characteristic to verify whether it would be possible to re-equilibrate the hyper into a lower methylation rate.

7. From an epigenome point, it would be interesting to get better knowledge on some histone marks at the 881 significantly differentially methylated CpG. This knowledge might be of interest for several reasons: i) useful hints of complex epigenome deregulation; ii) identification of targets that can be pharmacologically modulated.

8. From a bio-marker, medical point, which is the lowest discriminating number of differentially methylated CpG which could be used prognostically to discriminate JMML with poor from better prognosis?

Reviewer 1:

If gene expression would be used instead of methylation data, would similar clusters be identified? In other words: how different is methylation data from gene expression data for clustering of the patients?

We thank the reviewer for asking this important question and we have now addressed this. As the reviewers are well aware, the role that DNA methylation plays in regulating gene expression is complex. While early studies suggested that a direct relationship existed between CpG methylation (particularly in promotor regions) and the expression of the associated genes (reviewed in PMIDs 9395433, 10087932), recent studies favor a more complex interaction with other epigenetic phenomena, in particular, histone acetylation and histone methylation (reviewed in PMIDs 24555846, 26934913).

Similarly, early studies analyzing the effects of azacitidine in leukemia were based on the hypomethylating effects of the drug and re-expression of putative tumor suppressors that had been previously silenced (reviewed in PMID 12154409). Elegant recent experiments have led to a different hypothesis involving the immune system, and now the process of viral mimicry has taken hold as the leading theory of azacitidine's mechanism of action (PMID 26317465).

While the current prevailing hypotheses in the field of DNA methylation suggest that the influence of DNA methylation on gene expression is likely to be complex, it is still important to explore. We therefore generated RNA-Seq data from 8 of the 39 patients in our DNA methylation cohort who had RNA available; these 8 patients were evenly distributed among the three methylation clusters (4 in low methylation cluster, 2 in medium, 2 in high).

In our original DNA methylation analysis, patients clustered into three distinct groups based on the 1,527 most variable CpG sites. For this gene expression analysis that is described on line 198, we utilized the subset of the 1,527 CpG sites that are in the promoter region of a gene, and then analyzed the expression of those genes in the 8 JMML patients. As can be seen in supplemental figure 5, the clustering of JMML patients using this expression data is different than the clustering seen using DNA methylation data, suggesting that there is a complex relationship between these methylation changes and gene expression. This discussion has been added to the text starting on line 205.

After analyzing a small but representative subset of our JMML patients at the gene expression level, we believe that altered DNA methylation in JMML is functioning in a more complex manner than simply altering expression of associated genes. DNA methylation is used as a diagnostic tool (PMID 27503138) and predictive biomarker in several other malignancies (PMID 24081945) without a clear understanding of its mechanism. However, regardless of the lack of association between expression and methylation data in this clustering analysis, our genome-wide methylation analysis has found that DNA methylation in JMML is remarkably predictive of outcome.

On line 156 it is described a multivariable regression model was used. Could you specify more in detail the model that was used?

We used a multivariable logistic regression model for four-year EFS with methylation cluster group, somatic mutations, and age as predictors. We have modified the text (line 157) to clarify this point: “Furthermore, when a multivariable logistic regression model for four-year EFS was fit using only the three variables with a univariable $P < 0.01$ (methylation cluster group, somatic mutations, and age), the methylation cluster grouping retained statistical significance ($P = 0.032$) (Supplementary Table 5). The number of somatic mutations was also significant ($P = 0.018$), while age was not ($P = 0.12$).”

Patient data was collected from the EWOG-MDS consortium. Could you specify from which trials these data was?

Patients were enrolled in the prospective 98 and 2006 trials (www.clinicaltrials.gov: #NCT00047268, #NCT00662090) of the European Working Group of MDS in Childhood (EWOG-MDS). This is updated in the text on line 320. It should be noted that these trials are largely biology trials without a therapeutic intervention.

Which software/package/tool was used for the pathway analyses?

LIMMA analysis was performed using the Bioconductor package in R. An updated reference is now provided. KEGG analysis was performed using KEGG Mapper <http://www.genome.jp/kegg/mapper.html>.

This was updated in the methods section of the manuscript on line 429.

Reviewer 2:

1. A cohort of 79 JMML patients was compared with 22 healthy controls, using the set of 1,527 CpG sites defined by the discovery cohort, 27/79 JMML clustered closely to the 22 healthy controls (Fig. 3). Of these 27 patients, 14 (52%) experienced spontaneous resolution and only two (7%) experienced an event within 4 years. Which is the EFS of the other 9 patients that also cluster with the normal and correspond to 41%?

There are 27 JMML patients that cluster closest to the healthy controls as depicted in Figure 3. As the reviewer points out, 14 of these 27 patients experienced spontaneous resolution. Two of these 27 patients experienced an event within 4 years. None of the remaining 11/27 patients experienced an event within 4 years. Overall, the EFS for the 27 patients that clustered closest to the healthy controls was 93%. We are unaware of any other biomarker in JMML that similarly predicts such an outstanding event-free survival.

2. The authors carried out supervised analyses using the combined discovery and validation cohorts to identify CpG sites that differed between the 32 patients who experienced events and the 47 that did not experience events. 881 significantly differentially methylated CpG sites were defined. The authors retained only those CpG sites that are in gene promoters, yielding a list of 633 genes. On the contrary, if retaining those CpG sites that do not stand in promoters, where and how those sites could be grouped? Would also this signature be important for disease progression and development?

To address this issue and per both reviewers' suggestions we have incorporated RNASeq data into our analysis. As mentioned above, the regulation of gene expression by changes in DNA is complex. Several early studies (reviewed in PMIDs 9395433, 10087932) suggested a more direct role with more recent studies highlighting a more complex interplay of other epigenetic factors (reviewed in PMIDs 24555846, 26934913). Studies that have found a correlation, have frequently used an empiric approach where LIMMA analyses are performed on both DNA methylation and expression data, and then only focusing on overlapping genes to identify those most likely to be functionally influenced by changes in DNA methylation (PMID 26373278). We have taken a similar approach and have therefore updated our pathway analysis to reflect what occurs when we only use the overlapping genes between the two LIMMA analyses. We have also separately performed pathway analysis on genes that map to CpG probes in the DNA methylation specific LIMMA analysis.

In response to the original question, we now show the distribution of CpG sites relative to CpG islands and genes (Supplementary Figure 6). Nearly two-thirds of the CpG probes are found in promoter regions of genes as opposed to exonic, intronic or intergenic regions. Per the reviewer's suggestion, we ran pathway analysis including genes with CpGs in their exons or introns as well as in their promoters but we could not appreciate any meaningful differences versus only looking at genes with CpGs in their promoters.

In addition, KEGG pathway analysis of these same 633 genes revealed enrichment in several pathways, including MAPK, PI3K-AKT, Rap1 and Ras signaling, implicating these variably methylated CpG sites in leukemogenesis. Were the differential methylations at these sites influencing the corresponding gene expression? If so would these data indicate a potential for targeting MAPK, PI3K-AKT in a subset of JMML?

As discussed above, we now focus on genes that were found in the LIMMA analyses using both RNASeq and methylation data. Of note, the Ras pathway was the third most represented category in this analysis (line 222). We acknowledge that only a small number of genes overlap and are hesitant to overemphasize the relevance of this analysis.

Regardless of the role that DNA methylation plays in regulating activation of the Ras pathway, JMML has long been considered one of the purest Ras/MAPK driven diseases because virtually 100% of patients have an initiating mutation in the Ras pathway (PMID 26457647) that is always present upon relapse.

As the reviewer suggests this would be an ideal population in which to test inhibitors of the Ras pathway including its effector proteins. In fact, we have recently received approval from the Children's Oncology Group to test the MEK inhibitor trametinib in a trial for JMML patients with relapsed or refractory disease and plan to enroll patients by the end of 2017.

3. Pathway analysis confirmed that the most variably methylated CpG sites that were used in the supervised analysis were centered on genes involved in development and Ras/MAPK/PI3K/AKT signaling. Was the variation relative to common hyper or hypomethylation at these sites, or?

The number of CpG sites that were differentially methylated specific to these pathways in the overlapping LIMMA analyses were limited. For example, only two genes were identified in the Ras signaling pathway in our overlapping LIMMA analyses between DNA methylation and gene expression. In our opinion, this precludes a definitive interpretation about the direction of the relative hypo or hypermethylation at these sites.

4. The authors indicate that second mutations were found on some genes such as DNMT3. Which are the expression levels of those targets? Are they also affected by DNA hyper-hypomethylation? On a different corner, which is the mutated-DNMT3 enzymatic function Is it increasing or decreasing?

There are three patients that have secondary genetic mutations in epigenetic regulating genes including one patient harboring a *DNMT3A* p.G707fs mutation and two different patients with *ASXL1* mutations (p.E727* and p.H630fs) as shown in Supplementary Table 4. None of these patients had RNASeq data available for analysis to investigate changes in expression in these genes or their targets.

However, there is extensive data available regarding these mutations in other leukemias. The canonical hotspot (codon R882) in *DNMT3A* is assumed to have a dominant negative effect on methylation. There is evidence in the literature that the R882 mutations lead to global hypomethylation with a specific signature of hypermethylation at specific tumor suppressors (PMID 28288143). Similar to nearly all other pathogenic mutations outside of R882, our patient has a truncating frameshift mutation in *DNMT3A*. However, even in patients with *DNMT3A* mutations, the impact on leukemogenesis is still thought to be in part, independent of altered DNA methylation (PMID 27010239).

Similarly, nearly all pathogenic *ASXL1* mutations are inactivating in nature (PMID 22897849). Both patients in our cohort with *ASXL1* mutations have truncating mutations, consistent with the literature. As has been previously described, despite the

role of *ASXL1* predominantly affecting chromatin (PMID 22897849), nearly all patients with *ASXL1* mutations have hypermethylated DNA revealing the complex interplay between histone methylation/acetylation and DNA methylation (PMID 23066032). Both of the patients with *ASXL1* mutations in our cohort were categorized in the high methylation cluster.

5. The reason why the DNA methylation is high or low in the different patients, is not clear and not experimentally addressed. It would be good to tune this down or include more data suggesting potential hypothesis.

This is a very important point and we have revised the manuscript to tone down any suggestion that we have revealed an unknown mechanism for altered methylation in JMML. In particular, we have removed the limited GSEA and Gene Ontology analyses which did not contribute any additional information beyond the pathway analysis that could be mechanistically explained. As the reviewers are aware, the reason why DNA methylation is altered (hypo or hypermethylation) as a hallmark across all cancer types is still unknown. While a subset of acute myeloid leukemias have mutations in *DNMT3A*, *TET2* and *ASXL1* that explain a small portion of the altered methylome, the mechanism for the remaining majority of cancer types is still largely unknown.

6. The potential use of DNA demethylating agents should be investigated on primary cells or cell models with similar methylation characteristic to verify whether it would be possible to re-equilibrate the hyper into a lower methylation rate.

One limitation to studying JMML in general is the lack of any established cell lines owing to the myelodysplastic nature of the disease. While genetically engineered mice are available, these have murine methylomes which have not yet been characterized or compared to human samples. Also, primary JMML cells do not survive in culture long enough to measure DNA methylation after exposure to a demethylating agent like azacitidine. We are currently embarking upon JMML xenograft studies after recent work with collaborators has yielded consistently successful models (PMID 28576879) and will test azacitidine in those models but feel this experiment was outside the scope of this paper.

Despite all these limitations, this is a crucial question to answer. If hypomethylating agents are effective, that could open up new therapeutic opportunities outside of stem cell transplantation. Of note, azacitidine, the most widely used hypomethylating agent, is currently being tested in clinical trials across Europe in patients with JMML. However, it is not FDA approved for use in children in the United States. Our European colleagues who have access to the ongoing trial samples have recently published a manuscript that addresses this exact question. Their findings include an azacitidine induced hypomethylation effect. They also “specifically identified several differentially methylated coding and non-coding species of RNA, depicting a complex deregulation at different levels of transcription and translation in JMML.” Of note, this analysis was performed on

three JMML patients and we look forward to reading the results of their mature data in the coming years when the trial is completed.

7. From an epigenome point, it would be interesting to get better knowledge on some histone marks at the 881 significantly differentially methylated CpG. This knowledge might be of interest for several reasons: i) useful hints of complex epigenome deregulation; ii) identification of targets that can be pharmacologically modulated.

We agree entirely that analyzing the interplay between DNA methylation and histone marks is of broad interest and is a crucial piece of this observation and thank the reviewer for raising this point. Respectfully, we feel that these requested experiments are outside the scope of this current report but we plan on carrying out these exact experiments in the future.

8. From a bio-marker, medical point, which is the lowest discriminating number of differentially methylated CpG which could be used Prognostically to discriminate JMML with poor from better prognosis?

This is an excellent question and we have now amended the manuscript to describe how we plan on distilling the number of CpG sites so that this can be offered as a clinical test in the future. The most important classification in our discovery cohort was into one of three methylation sample clusters. This clustering was remarkably consistent when testing ~10,000, ~1000 and even ~100 of the most variable probes. Therefore, the question remains, what is the smallest number of probes that are necessary to recapitulate the same clustering. For this purpose, we used the discovery cohort to rank probes and the cluster call of the validation cohort to assess accuracy. We ranked probes from most typical to least typical on the discovery cohort, where the determination of “typical” was based on the distance from the sample to the centroid of the corresponding cluster summed over all samples. We then tested the calls on the validation cohort using from 5 to 200 probes, adding the best probes 5 at a time, with distance to the training set centroid the basis of the call. Disagreements with the calls based on all 1,527 probes were considered errors. As can be seen in Supplementary Figure 4, once we reached the level of 50 probes, the number of errors went down to either 0 or 1. In summary, 50 probes are sufficient for discrimination. This is a crucial first step in preparation for a clinical test for patients with JMML. We plan to collaborate with our colleagues from Germany who have co-submitted their article to eventually pool our combined methylation data and design a sequencing based assay for clinical implementation. This discussion has been added to the text starting on line 197.

REVIEWERS' COMMENTS:

Reviewer #1 (Remarks to the Author):

The authors have addressed the questions and comments of the reviewers and have incorporated some changes and additional experiments in the manuscript. This has improved the manuscript and I have no further comments.

Reviewer #2 (Remarks to the Author):

The revised version of the manuscript has improved addressing the majority of this reviewer concerns. This reviewer also recognizes that some of the suggestions were not addressed now due to the lack of reliable models and patient's material. This is well explained by the authors and looks understandable.

The fact that the authors plan to collaborate with the colleagues from Germany who have co-submitted a back to back article to pool their combined methylation data and design a sequencing based assay for clinical implementation is a very important point.

This may have a very relevant readout in the clinics as it will have a potential eli-combo treatment of some of those patients.

The discussion added to the text starting on line 197 might even more emphasise this point.

Reviewer #1 (Remarks to the Author):

The authors have addressed the questions and comments of the reviewers and have incorporated some changes and additional experiments in the manuscript. This has improved the manuscript and I have no further comments.

Reviewer #2 (Remarks to the Author):

The fact that the authors plan to collaborate with the colleagues from Germany who have co-submitted a back to back article to pool their combined methylation data and design a sequencing based assay for clinical implementation is a very important point. This may have a very relevant readout in the clinics as it will have a potential combo treatment of some of those patients.

The discussion added to the text starting on line 197 might even more emphasise this point.

We have now added another sentence to the discussion on line 331 which summarizes our plan to implement a clinical test by pooling our data with our European colleagues and offering a sequencing based DNA methylation assay across multiple continents.